# Modeling the Ecological Response of a Temporarily Summer-Stratified Lake to Extreme Heatwaves

**Weiyu Chen** [1,2,*], **Anders Nielsen** [2], **Tobias Kuhlmann Andersen** [2,3], **Fenjuan Hu** [2], **Qingchuan Chou** [2,4,5], **Martin Søndergaard** [2], **Erik Jeppesen** [2,3] **and Dennis Trolle** [2,3]

1   College of Water Conservancy and Hydropower Engineering, Hohai University, Nanjing 210098, China
2   Department of Bioscience, Aarhus University, Vejlsøvej 25, 8600 Silkeborg, Denmark; an@bios.au.dk (A.N.); tka@bios.au.dk (T.K.A.); fenjuan@bios.au.dk (F.H.); chouqc@ihb.ac.cn (Q.C.); ms@bios.au.dk (M.S.); ej@bios.au.dk (E.J.); trolle@bios.au.dk (D.T.)
3   Sino-Danish Centre for Education and Research, University of Chinese Academy of Sciences, Beijing 100049, China
4   Institute of Hydrobiology, The Chinese Academy of Sciences, Wuhan 430072, China
5   College of Life Sciences, University of Chinese Academy of Sciences, Beijing 100049, China
*   Correspondence: weiyuchen92@hotmail.com

**Abstract:** Climate extremes, which are steadily increasing in frequency, can have detrimental consequences for lake ecosystems. We used a state-of-the-art, one-dimensional, hydrodynamic-ecosystem model [General Ocean Turbulence Model (GOTM)-framework for aquatic biogeochemical models (FABM)-PCLake] to determine the influence of extreme climate events on a temperate and temporarily summer stratified lake (Lake Bryrup, Denmark). The model was calibrated (eight years data) and validated (two years data), and the modeled variables generally showed good agreement with observations. Then, a span of extreme warming scenarios was designed based on weather data from the heatwave seen over northern Europe in May–July 2018, mimicking situations of extreme warming returning every year, every three years, and every five years in summer and all year round, respectively. We found only modest impacts of the extreme climate events on nutrient levels, which in some scenarios decreased slightly when looking at the annual mean. The most significant impacts were found for phytoplankton, where summer average chlorophyll a concentrations and cyanobacteria biomass peaks were up to 39% and 58% higher than during baseline, respectively. As a result, the phytoplankton to nutrient ratios increased during the heat wave experiments, reflecting an increased productivity and an increased cycling of nutrients in the pelagic. The phytoplankton blooms occurred up to 15 days earlier and lasted for up to half a month longer during heat wave years relative to the baseline. Our extreme scenarios illustrated and quantified the large impacts of a past heat wave (observed 2018) and may be indicative of the future for many temperate lakes.

**Keywords:** extreme climate; GOTM-FABM-PCLake; temporary stratified lake; lake ecosystem; heatwave 2018

## 1. Introduction

The global average surface temperature has increased by approximately 0.6 °C in the past century [1], and the Intergovernmental Panel on Climate Change (IPCC) predicts increases in global surface temperature of 1.4–5.8 °C for the year 2100 [1]. Especially in 2018, the extreme summer temperature occurred in Europe, parts of North America, and East Asia. Denmark had an extremely warm, long lasting summer compared with those in the past few decades [2]. In 2018, Danish summer season showed record warmth with a record-breaking number of summer days and the highest low temperatures since 1874. Such changes are expected to affect lake ecosystem dynamics by altering the

trophic structure and interacting with nutrients, potentially intensifying eutrophication and human health problems, the latter due to enhanced growth of potentially toxin producing cyanobacteria [3,4].

Global warming is also expected to result in a higher frequency and magnitude of climate extremes, such as severe droughts, heavy rainfall, and heat waves (IPCC 2014) [5]. It was recently shown that the European summer might experience a marked increase in year-to-year temperature variations and incidences of heat waves [6,7], which may pose a serious threat to lake ecosystems. Shallow lakes, having a strong interaction between sediment and the overlying water, are highly sensitive to climate change [8]. With inclining temperatures, the current pattern of temporary lake stratification may change and occur more frequently with longer duration in shallow lakes, creating a more complex nutrient and ecological dynamic. Temporary stratification may accelerate release of nutrients from the sediment due to oxygen depletion near the bottom, and these nutrients may support the growth of phytoplankton when water mixes again. Climate change is also projected to result in increases in precipitation, which may change the hydrology of lake catchments [9] and lead to higher export of nutrients from watersheds to north European lakes, further enhancing eutrophication [10]. Extreme and uncommon events such as this pose abrupt and significant risks to the environment and society, which draws attention to the need to further understand its impact on lakes.

Several methods can be used to estimate the response of lake ecosystems to climate change [11,12]. These include analysis of long-term data series from monitoring, space-for-time analysis of data collected across latitudes, paleolimnology, experimentation (e.g., mesocosms), and modeling. Process-based ecological models can dynamically simulate the complex responses to changes in climate as they attempt to account for temperature effects on a wide range of processes. However, the value of such models depends on their accuracy in simulating the ecosystem and their ability to adapt to the observed change at a scale of interest to scientists or managers.

In this study, we aimed to determine the ecosystem response to extreme climate events for a temporarily stratified, eutrophic lake (Lake Bryrup, Silkeborg, Denmark) using a state-of-the-art and comprehensive ecosystem model [General Ocean Turbulence Model (GOTM)-framework for aquatic biogeochemical models (FABM)-PCLake] and air temperature data from the unusually warm summer of 2018 as a basis for the extreme scenario analysis. We hypothesized that summer stratification would be stronger and of longer duration during a heat wave year, leading to increases in the internal loading of nutrients. This would stimulate phytoplankton growth, which together with the elevated temperatures would lead to higher dominance of cyanobacteria.

## 2. Methods

### 2.1. Study Site

Lake Bryrup is shallow lake located in the Central Region of Denmark, southeast of the town of Silkeborg (56.02° N, 9.53° E). The lake has a mean depth of 4.6 m and a maximum depth of 9 m. Its surface area is 0.37 km$^2$, the lake volume is 1.72 million m$^3$, and the hydraulic retention time varies between 2–3 months. The average inflow discharge is about 0.25 m$^3 \cdot$s$^{-1}$, and the outflow discharge almost equals the inflow discharge. The lake temporarily stratifies during summers. During 1996–2005, the average chlorophyll a (Chl a) concentration in the lake was 26.3 µg$\cdot$L$^{-1}$, which is eutrophic, and average total nitrogen (TN) and total phosphorus (TP) concentrations were 3.37 mg$\cdot$L$^{-1}$ and 0.094 mg$\cdot$L$^{-1}$, respectively. Most of the water comes from the stream Nimdrup Bæk in the south, and the water flows from the lakes Kvindsø and Kulsø to the stream Salten Å and the River Gudenå (Figure 1). The watershed is 49.9 km$^2$, of which 60% is used for agriculture, 12% is used for forest, and about 10% is used for urban area. Lake Bryrup receives a large amount of nutrients via inlets that drain agricultural areas and small towns in the countryside, which has resulted in an increased algae growth and cyanobacteria blooms during summer months. Measures (notably improved sewage treatment and diversion of part of the sewage, reduction of diffuse sources) have been taken to reduce the external nutrient loading and, as a consequence, TN loading has decreased from 8.1 to 5.7 g$\cdot$m$^{-2}$ lake area$\cdot$year$^{-1}$

and TP loading from 0.15 to 0.08 g·m$^{-2}$·year$^{-1}$ during 1990–2014. The effect of the measures has been positive, especially for lake TP, which has declined by almost 50% from approximately 0.15 mg·L$^{-1}$ to 0.10 mg·L$^{-1}$.

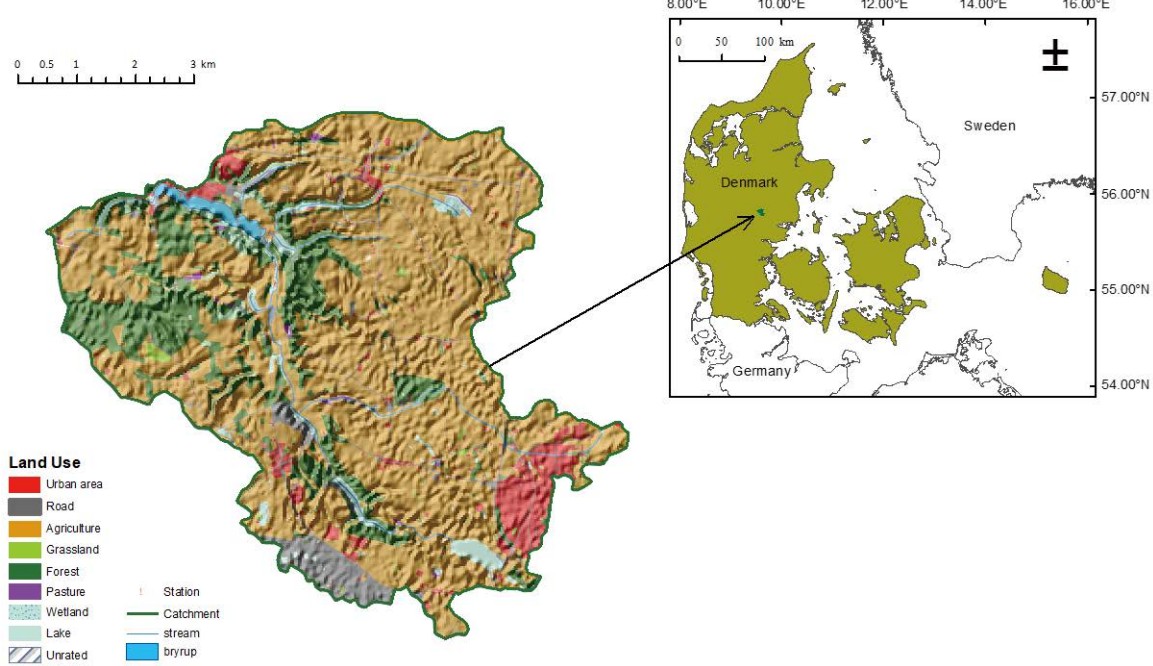

**Figure 1.** Map of the study lake, Lake Bryrup, and its watershed.

## 2.2. Model Description

For simulating the lake ecosystem, we used the complex GOTM-FABM-PCLake model (https://projects.au.dk/wet/), a process-based, two-way coupled hydrodynamic-ecosystem model. It consists of a one-dimensional hydrodynamic model (GOTM) coupled with a lake ecosystem model (PCLake) via the framework for aquatic biogeochemical models (FABM). The model complex describes the most significant hydrodynamic, biotic, and abiotic components of an aquatic ecosystem. The GOTM-lake is a water column model of the principal hydrodynamic and thermodynamic processes related to vertical mixing in natural waters [13,14]. The chemical and the biological dynamics in the water column and the sediment are based on a further development of the PCLake model (for more details, see Janse 2005 [15,16]), namely FABM-PCLake by Hu et al. 2016 [17]. The state variables of stratified lakes are different at different layers due to the influence of momentum and heat. The variables simulated by the biochemical model are transported to the hydrodynamic model GOTM through FABM, then the hydrodynamic model integrates ecological and mixing processes and sends updated states back to the biochemical model [17]. The GOTM-FABM-PCLake ecosystem model complex differs from PCLake by taking into account: (i) the stratification over the water column for several state variables (temperature, nutrients, phytoplankton, etc.); (ii) more complex description of sediment nutrient dynamics, e.g., with regards to release of nutrients by including a physically based sediment resuspension routine.

## 2.3. Model Input Data

Model input files include data on the physical domain of the lake (hypsography), flow discharge, inflow nutrient concentration, and meteorological forcing (Table 1). Since GOTM is a vertical hydrodynamic model, the physical domain is specified as a hypsography, which represents the relationship between depth (m) and the corresponding horizontal area (m$^2$) for a given water layer, which is easily obtained from bathymetric maps. Inflow discharge (m$^3$·s$^{-1}$) and inflow nutrient concentrations, including nitrate (NO$_3$) (mg N·L$^{-1}$), ammonia (NH$_4$) (mg N·L$^{-1}$), dissolved organic

nitrogen (mg N·L$^{-1}$), particulate organic nitrogen (mg N·L$^{-1}$), phosphate (PO$_4$) (mg P·L$^{-1}$), dissolved organic phosphorus (mg P·L$^{-1}$), and particulate organic phosphorus (mg P·L$^{-1}$), are also required as boundary conditions. Discharge and concentrations of nitrogen (N) and phosphorus (P) were obtained from the National Monitoring Program (NOVANA) [18]. Daily meteorological data input to the model include wind speed in both N–S and E–W directions (m·s$^{-1}$), air pressure (hPa), air temperature (2 m height) (°C), dew-point temperature (°C), and cloud cover fraction (varying from 0–1). These data were obtained from the ERA-Interim reanalysis dataset by the European Center for Medium-Range Weather Forecasts (ECMWF) [19], which is a consistent and continuous 3-hour data series.

**Table 1.** The input and some selected output of the model.

| | Input | Output |
|---|---|---|
| Physical domain | Longitude, latitude, depth (m), and the corresponding horizontal area (m$^2$) for a given water layer (hypsography) | |
| Flow discharge | Inflow discharge (m$^3$·s$^{-1}$), outflow discharge (m$^3$·s$^{-1}$) | |
| Inflow nutrient concentrations | NO$_3$ (mg N·L$^{-1}$), NH$_4$ (mg N·L$^{-1}$), dissolved organic nitrogen (mg N·L$^{-1}$), particulate organic nitrogen (mg N·L$^{-1}$), PO$_4$ (mg P·L$^{-1}$), dissolved organic phosphorus (mg P·L$^{-1}$), particulate organic phosphorus (mg P·L$^{-1}$) | Temperature, DO, NO$_3$, NH$_4$, TN, PO$_4$, TP, Chl a |
| Meteorological forcing | Wind speed in both N–S and E–W directions (m·s$^{-1}$), air pressure (hPa), air temperature (2 m height) (°C), dew-point temperature (°C), and cloud cover fraction (varying from 0–1). | |

DO, dissolved oxygen. NO$_3$, nitrate. NH$_4$, ammonia. TN, total nitrogen. PO$_4$, phosphate. TP, total phosphate. Chl a, chlorophyll a.

## 2.4. Model Calibration and Validation

Currently, the most complete dataset available for Lake Bryrup covers the period 1990–2005. To reduce the uncertainty associated with model initialization in relation to model calibration, the years 1990–1995 were used as a model warm-up period. The model was then calibrated against observed data covering the following eight-year period (1996–2003) for variables of temperature, dissolved oxygen (DO), NO$_3$, NH$_4$, TN, PO$_4$, TP, and phytoplankton concentrations (represented by Chl a). The model's initial calibrated parameters and boundary conditions were based on predefined values in the GOTM-FABM-PCLake model and site-specific measurements for Lake Bryrup. On the basis of a sensitivity analysis previously performed by Janse et al. [20] as well as model experience, 117 parameters were subject to adjustment to achieve feasible model performance of the water quality variables (see Appendix A). The model was calibrated using ACPy, a Python based auto-calibration tool developed for GOTM-FABM-PCLake. ACPy uses a differential evolution algorithm to calibrate model parameters by optimizing against a maximum likelihood function. Calibration was performed through a series of steps, analogue to the methodology by Trolle et al., where each step targets a selection of parameters identified to have strong influence on temporal and vertical dynamics of specific state variables of which observations exist for the lake. In each step, several calibration iterations were conducted, switching between adjustments of the parameter ranges and evaluation of model performance through root-mean-square-error (RMSE) and the coefficients of determination ($R^2$) between model output variables and observed water quality data [21]. The model calibration in each step continued until there was little advancement in RMSE and $R^2$ values. Then, model validation was estimated for a two-year period (2004–2005) for all variables with corresponding RMSE and $R^2$ values calculated.

*2.5. Base Scenario and Extreme Climate Scenarios*

From our calibrated base model simulation, we designed one baseline and a total of five scenarios—three scenarios with real world extreme weather events (summer 2018) and two with generated extreme weather events, all running for 50 years to project the climate effects on lake ecosystem. The baseline represented loading and meteorology data from 2001 to 2005 looped forward (as five-year periods) until 2050. The extreme weather event in year 2018 was then applied to the baseline to quantify the impacts of extreme weather on the lake ecosystem. In the first three scenarios, we swapped (with varying frequency) each 3-hour air temperature record from May to end of July with the equivalent record (match on day and time) of air temperature from the 2018 summer data—every five years (SC1), every three years (SC2), and every year (SC3). For the last two scenarios, a long-term delta change was created based on a temperature difference analysis. Here, we calculated the mean air temperature difference for each day and year (2001–2005) in a particular year compared to temperatures in the summer 2018. Subsequently, the means of quantile 50 (2.9 °C) (SC4) and 75 (5.0 °C) (SC5) were applied to all 3-hour records separately after 2005. We extracted and analyzed the predicted variables of the last five years (2046–2050) of the 50-year scenario period.

## 3. Results

*3.1. Model Calibration and Validation*

The modeled temperature corresponded well with observed values (Figure 2). For oxygen, the model succeeded in capturing the seasonal trends in concentrations; however, some steep short-term decreases during winter were not captured well during calibration or for the validation period.

As $NO_3$ constitutes the main part of TN, TN concentrations showed the same trend as $NO_3$ concentrations. The modeled TN and $NO_3$ generally tracked annual and seasonal dynamics of the monitoring data. However, modeled $NO_3$ occasionally missed maxima and minima (e.g., 1998; Figure 2), and so did TN. While the modeled $NH_4$ successfully captured trends in concentrations during the entire period 1996–2005, the monitoring data in the summer were underestimated, particularly at some high peaks. $NH_4$ concentrations in the epilimnion showed a bit of a poor fit, and the model lacked the capability to simulate seasonal peak dynamics in both the epilimnion and the hypolimnion. $NO_3$ and $NH_4$ had only one peak per year, in spring and in autumn, respectively (Figure 2). Then, $NH_4$ concentrations started to increase in summer, mainly at the bottom of the water column.

There was good agreement between the model output and the monitoring data for TP. However, the model at times tended to underestimate the observed TP, especially for the hypolimnion, and it did not capture the full extent of the high summer peaks. The $PO_4$ concentrations were rather poorly reproduced for the summer period, especially in the hypolimnion, the monitored high peaks in summer being overestimated (Figure 2). Moreover, the predicted winter peaks of $PO_4$ were slightly higher than those recorded.

The dynamics of total Chl a were reproduced reasonably well, capturing seasonal and inter-annual variability of the monitoring data. However, the simulated autumn blooms were occasionally underestimated (e.g., 1998), while the spring bloom peaks were overestimated for some years, especially during the calibration period.

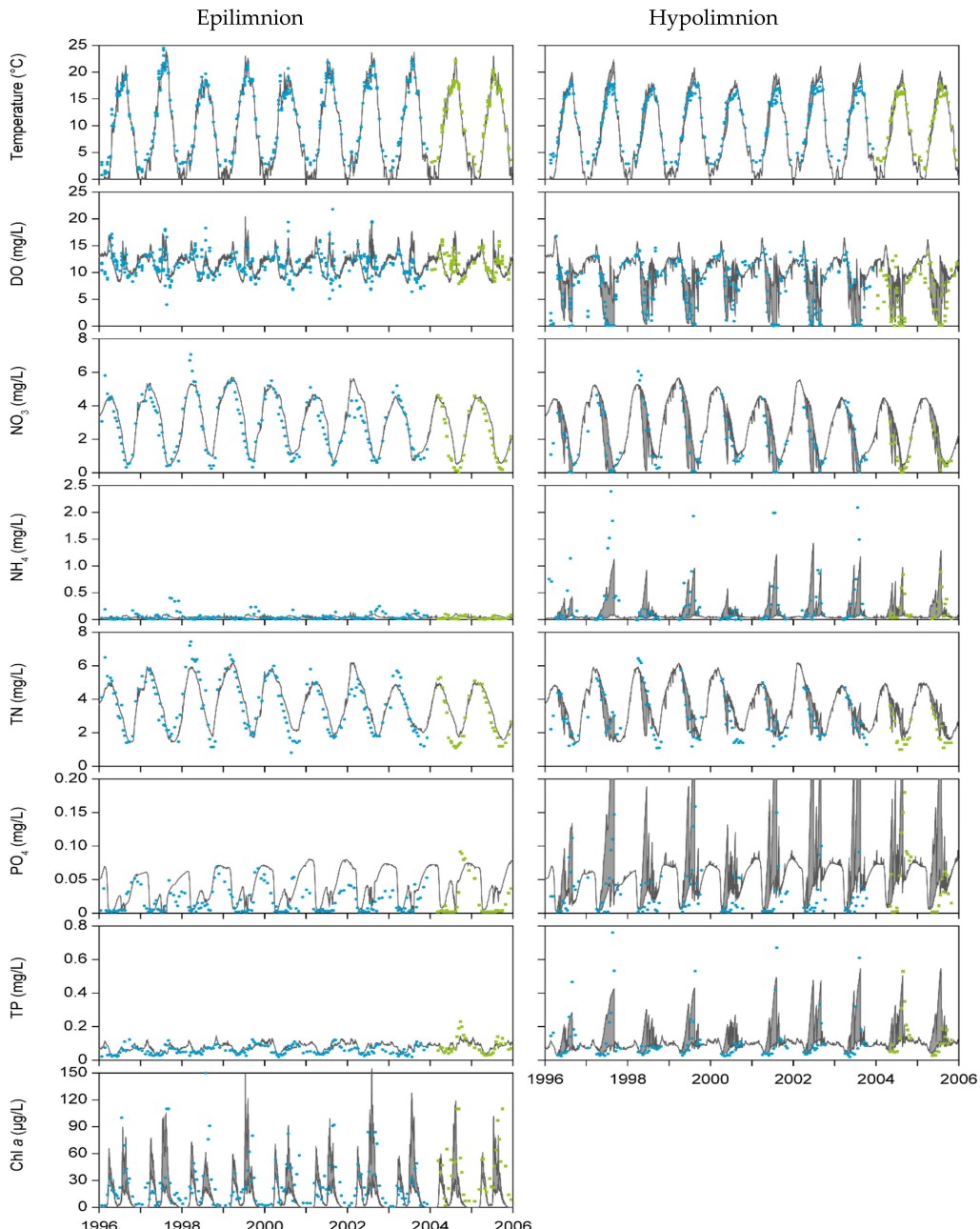

**Figure 2.** Comparison of model simulation results (line) against observations (dots) during the calibration (blue circles) and the validation periods (green squares) in the epilimnion (0.0––3.0 m) and the hypolimnion (−6.0––10.0 m), respectively. A, temperature. B, DO. C, nitrate. D, ammonium. E, total nitrogen. F, phosphate. G, total phosphorus. H, Chl a.

## 3.2. Extreme Climate Scenarios

Relative to baseline, the model predicted, as expected from the changes in air temperature, a gradual increase in water temperature from SC1 to SC5 (Figure 3). Temperatures in scenarios SC3–SC5 increased by 6% to 17% relative to the baseline. For the simulated TN concentrations, which gradually declined with increased air temperature, the effects of extreme warming in SC1 and SC2 were similar in the surface water, and there was an evident decrease in TN concentrations from SC3 to SC5, from 5% to 13%. TP concentrations in the extreme climate effects scenario were somewhat lower than those in baseline, and the most marked changes in simulated TP occurred in SC3 and SC5 with a decrease of 3% and 3.5%, respectively. The impacts of extreme warming in scenarios SC1–SC5 were most pronounced

for Chl a concentrations and cyanobacteria biomass, with a considerable increase during summer. Otherwise, the increases in the dynamics of Chl a and cyanobacteria were similar, starting with a slow increase in SC1 and SC2 followed by a sharp increase in SC3–SC5 (Figure 3). As in Lake Bryrup, the phytoplankton composition during summer was dominated by cyanobacteria, and the projected phytoplankton biomass approximately resembled the pattern of cyanobacteria biomass (Figure 4). In the last year of scenarios (2050), the bloom timing of cyanobacteria in SC5 was 15 days earlier compared to baseline.

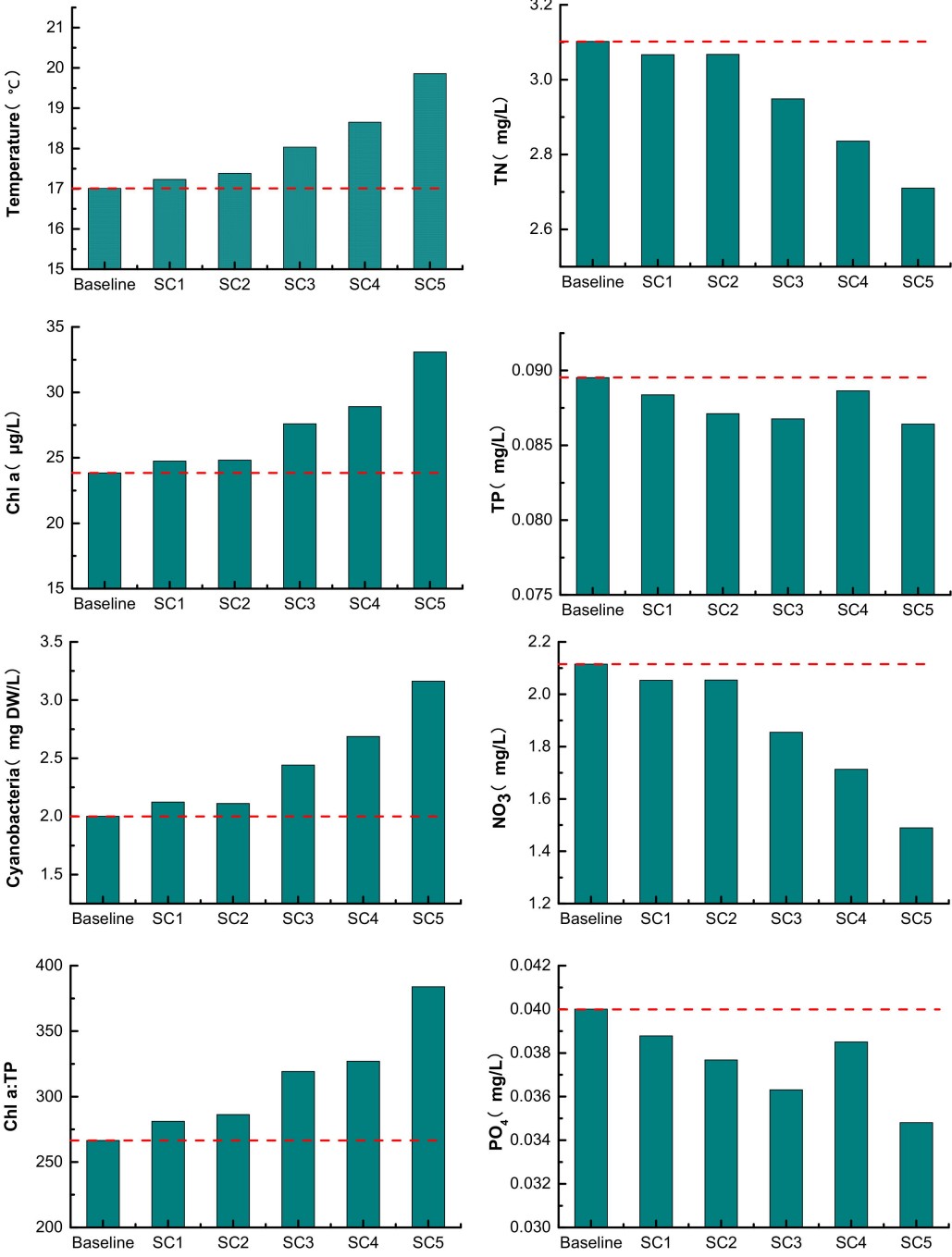

**Figure 3.** Simulated effects of extreme summer weather on water temperature, TN, TP, Chl a concentration, cyanobacteria biomass, $NO_3$, Chl a:TP ratio, and $PO_4$ in the surface of Lake Bryrup based on summer averages from daily values during the last five years.

In baseline, the period where the lake is subject to temporary stratification events lasted about four months, from May to August, and often there was full mixing of the water column from bottom to surface in June every year (Figure 5). The summer stratification of the lake was strengthened in the extreme climate scenarios. For nitrogen concentrations, the scenario results (Figure 3) demonstrated minor impacts of extreme warming on TN and major impacts on inorganic nutrients. The most notable changes in $NO_3$ and TN in the future scenarios were a 77% and a 13% decrease, respectively, both in SC5. The longer duration of cyanobacteria blooms (Figure 4) in the extreme warming scenarios implied higher $PO_4$ consumption and lower TP than in baseline. The most pronounced impact of warming occurred in SC5, with TP and $PO_4$ concentrations decreased by 3.5% and 13%, respectively, compared with baseline. With increasing temperature, the model predicted progressively more severe summer blooms and increasing Chl a:TP, as also seen in empirical studies [22,23]. Due to the impact of extreme warming, the duration of summer blooms was longer due to an earlier (around two weeks) onset of the blooms (Figure 4). Moreover, the intensity of both the spring and the summer blooms was greater, and in SC5, the peak concentration of Chl a was 46% higher than in baseline (Figure 4).

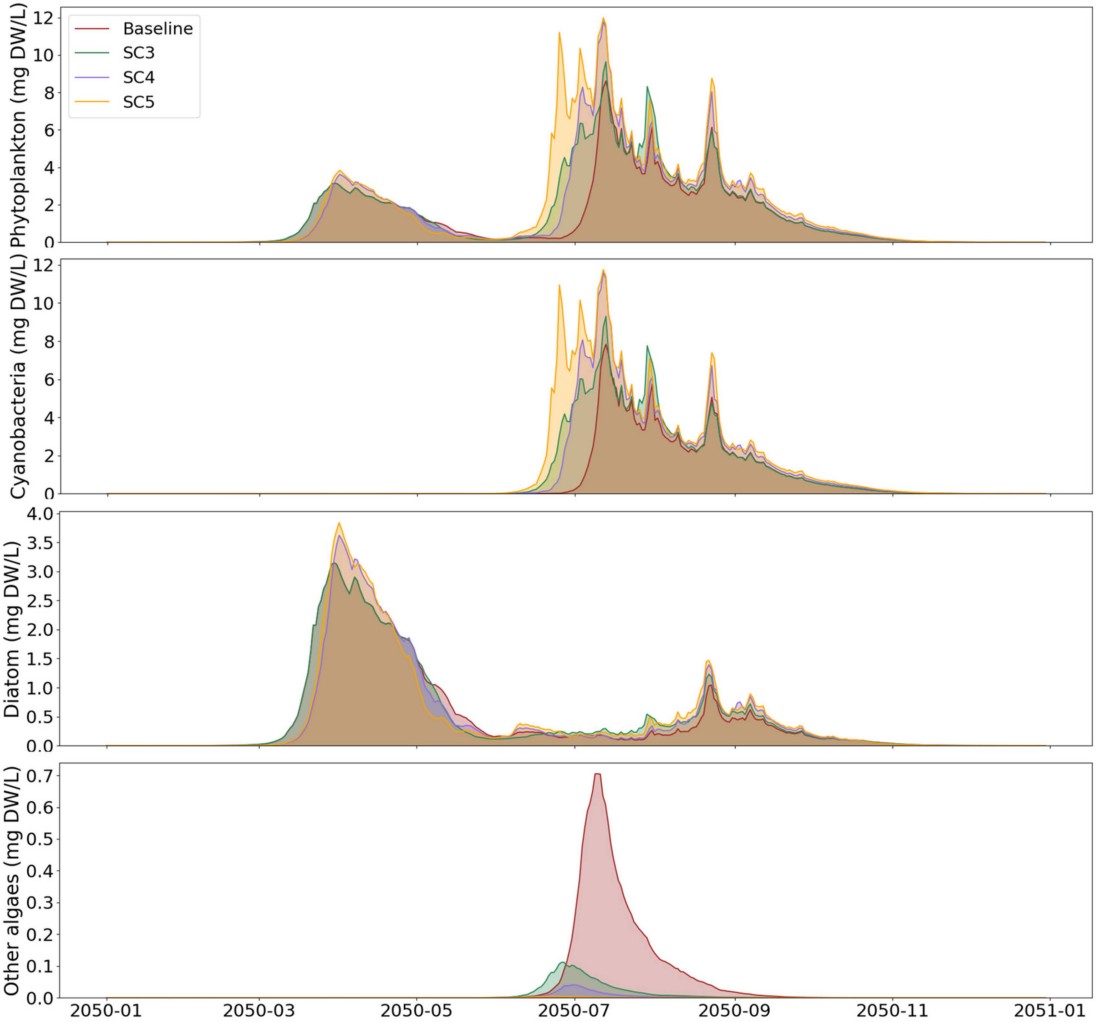

**Figure 4.** Simulated phytoplankton, cyanobacteria, diatom, and other algae biomass of baseline, SC3, SC4, and SC5 during the last simulation year (2050).

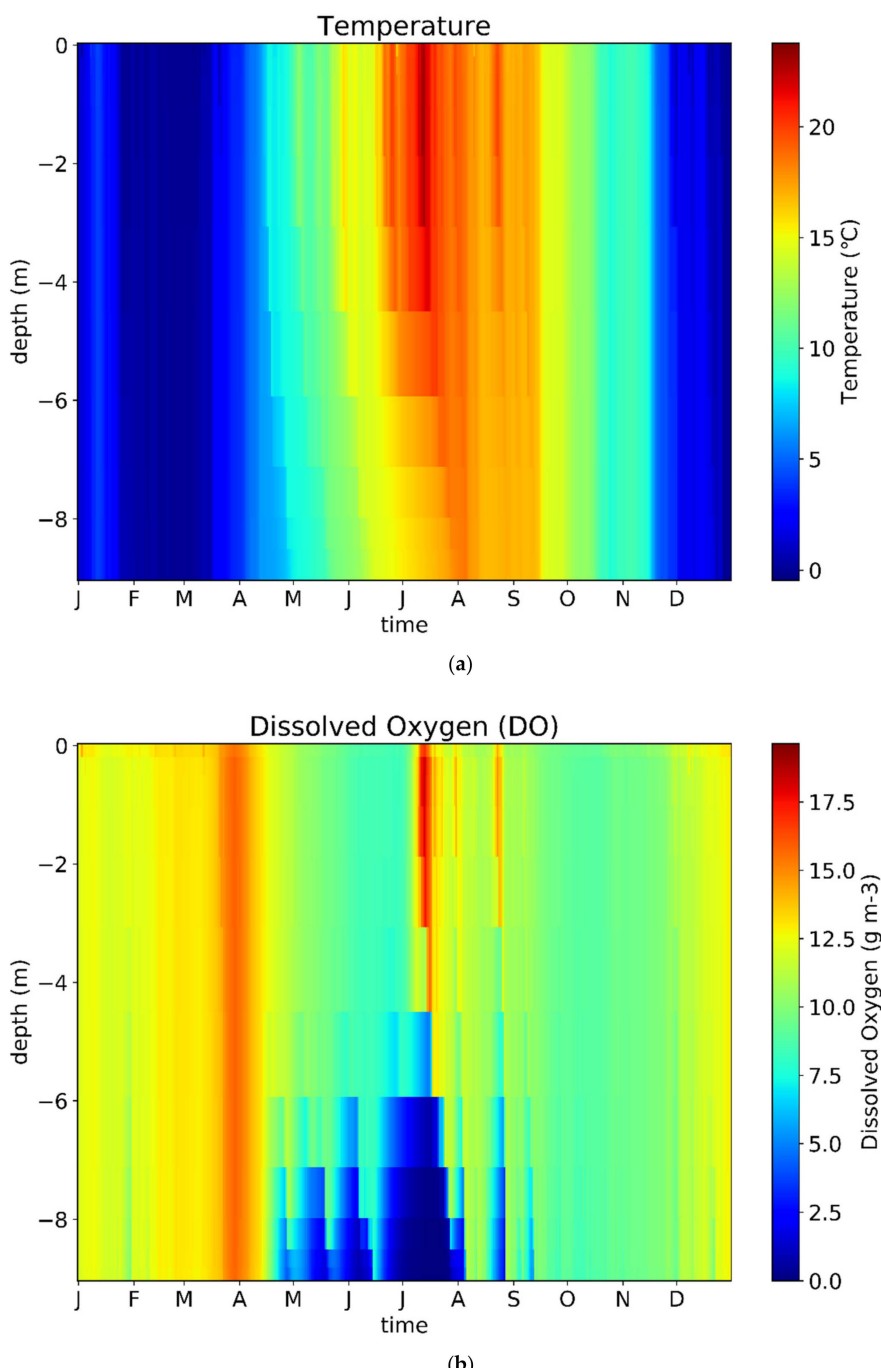

**Figure 5.** Simulation for the whole water column in 2050 in base scenario. (**a**) Temperature; (**b**) DO.

## 4. Discussion

### 4.1. Model Performance

The GOTM-FABM-PCLake model reproduced satisfactorily the seasonal dynamics for most state variables of Lake Bryrup although some discrepancies between model and observations occurred (Table 2). Conceptually, the model accounts for key ecological processes and predicted the impact of extreme warming on lake trophic components, including Chl a, DO, $NO_3$, $NH_4$, TN, $PO_4$, TP, and water temperature. However, due to the large variability of observations, the projections should still be interpreted with some caution.

**Table 2.** Root-mean-square-error (RMSE) values and coefficients of determination ($R^2$) between simulated model output and observations separated in calibration and validation.

| | $R^2$ | | RMSE | |
|---|---|---|---|---|
| | Calib. | Valid. | Calib. | Valid. |
| Temperature | 0.98 | 0.98 | 1.37 | 1.41 |
| DO | 0.45 | 0.38 | 3.09 | 3.15 |
| $NO_3$ | 0.85 | 0.85 | 0.69 | 0.71 |
| $NH_4$ | 0.45 | 0.51 | 0.32 | 0.15 |
| TN | 0.79 | 0.81 | 0.71 | 0.78 |
| $PO_4$ | 0.30 | 0.19 | 0.03 | 0.04 |
| TP | 0.31 | 0.15 | 0.08 | 0.08 |
| Chl a | 0.29 | 0.28 | 23.65 | 35.91 |

Note: Calib. = calibration value; Valid. = validation value.

In general, the coefficients of determination ($R^2$) between the modeled and the observed data demonstrated a good fit for temperature (Table 2); however, the observed temperature in winter was underestimated by the model. This may reflect that ice cover is not yet explicitly included in the GOTM-FABM-PCLake model and, furthermore, we did not account for the impact of streams, resulting in a lower modeled winter lake temperature.

The modeled oxygen concentrations overall exhibited the same variations as the measured data during calibration and validation periods, except periodically in winter. The model generally showed a lack in accuracy when the temperature was low.

The model performance for nitrogen was generally better than in most other lake model studies [24–26]. However, the $R^2$-values of inorganic nutrients showed a slightly poorer fit, such as $NH_4$. $NH_4$ is labile and affected by phytoplankton uptake (included in the model) and nitrification. In the epilimnion, the consumption of $NH_4$ as the main nutrient for cyanobacteria is large in summer. The denitrification and the nitrification, which are important for $NH_4$ variation, are affected by oxygen concentration. In this study, the oxygen changed fast when the lake became fully mixing (Figure 4), therefore $NH_4$ dynamics showed quick changes in the surface, which may explain the poor fit. Periodic resuspension events caused by wind-induced shear stress and bioturbation may significantly affect the exchange of sediment nutrient, including nitrogen, and enhance denitrification [27,28]. Therefore, the relatively simplified and empirical-based conceptualization of resuspension of this model might have influenced the denitrification levels and thus inorganic N.

The modeled phosphorus concentrations generally exhibited the same seasonal dynamics and inter-annual variations as the measured data during the calibration and the validation periods. The summer $PO_4$ peak reflects internal factors (decomposition and nutrient release from the sediment), while variations and peaking in winter indicate that $PO_4$ was mainly due to external nutrient inputs combined with slow in-lake uptake due to low temperatures. The simulated too low nitrate concentrations in the bottom water layer before the full mixing of the water column (around July) may have resulted in a reduced release of $PO_4$ compared with the observations. Nitrate is part of the diagenetic sequence in lake sediments and affects the redox potential, as has been seen in field studies in several lakes [25,29]. Then, because of the high nutrient consumption during severe summer blooms followed by sedimentation, the available amount of $PO_4$ did not suffice for diatom development in spring, showing a bloom delay in the last two high temperature scenarios. In addition, the simplified process of sediment diagenesis might have contributed as well. For example, decomposition of sediment layers and phosphorus release induced by high pH are not included in the model [14]. These factors have previously been shown to affect phosphorus release, especially in eutrophic lakes in the summer [30].

For Chl a, the model generally showed good agreement with the field records mimicking the two peaks per year. Our model included three algae forms: cyanobacteria, diatoms, and "other algae". Diatoms were the dominant algae in the spring, while cyanobacteria dominated in the summer, reflecting that the optimal growth temperature of the algae are different. According to the calibration,

the optimum temperature for diatom growth (20.3 °C) was lower than that for cyanobacteria (28.1 °C); thus, the conditions for diatom growth were most favorable in spring and for cyanobacteria in late summer.

### 4.2. Effects of Extreme Temperatures

In the five extreme temperature scenarios, summer Chl a concentrations increased gradually, and cyanobacteria contributed progressively more to the overall phytoplankton biomass, as seen in other model studies of climate warming effects [11,31]. Because the last two scenarios were applied for the whole year instead of only summer, the intensity of the diatom bloom was greater compared with baseline and the first three scenarios. Therefore, with the warmer climate, extensive surface $NO_3$ and $PO_4$ consumption occurred in summer, reflected in increasing phytoplankton biomass, especially cyanobacteria. Since $NO_3$ and $PO_4$ are the main components of TN and TP, respectively, the dynamics trend of TN and TP changed with the variations of $NO_3$ and $PO_4$. The first three scenarios (SC1–SC3) were applied only in the summer, and for the last two scenarios (SC4 and SC5), warming was applied year-round. Sediment nutrient release mechanisms and remineralization are related to the temperature [32]. Thus, compared with the SC1–SC3, more phosphorous under SC4 and SC5 could potentially be released from the sediment [33] not only in summer but also in spring and in autumn. Thus, phosphorus concentrations in SC4 were also overall slightly higher than those in SC3.

In general, we did not anticipate that nutrient levels would remain somewhat similar or even decrease in our heat wave experiments, as other studies found the opposite [12]. Previous studies applied somewhat simpler models; however, in those studies, for example, sediment nutrient pools were not dynamically accounted for. Model simulations in our study indicate that, although the periods where Lake Bryrup undergoes temporary stratification may be stronger and last a bit longer during a heat wave, the regeneration of sediment-bound phosphorus in-between stratification events is not sufficient to result in a marked increase in overall annual TP levels. Hence, we found that the initial sediment release of $PO_4$ during the first stratification event of a heat wave year would cause higher peaks of bottom water $PO_4$ and TP levels relative to baseline, but that the peaks of the following stratification events were somewhat lower compared to baseline.

### 4.3. Study Constraints

In this study, most of the selected model parameters and ranges used for calibration were derived from previous research using the PCLake model [11,34], while the rest were chosen on the basis of experience. Therefore, the final model parameter selection, among more than 300 parameters, may not be entirely accurate for the calibration of the model specifically for Lake Bryrup. In addition, a highly parameterized model such as GOTM-FABM-PCLake is also subject to some degree of equifinality, meaning that more than one set of parameter combinations could result in an overall similar and acceptable model performance during calibration. While we ignored this aspect in our study, it perhaps could potentially be accounted for by utilizing more than one (plausible) model parameterization during the scenario simulations, which then might also give insight into the potential uncertainty of the scenarios by providing an envelope of simulated outcomes rather than a single outcome of each scenario.

Although the GOTM-FABM-PCLake model is a state-of-the-art, one-dimensional lake ecosystem model, it still simplifies some aspects of a real lake ecosystem. The application of this model to Lake Bryrup was effective, but some model compartments can still be further developed to improve the performance, such as an explicit ice-cover module that caused the discrepancies seen for temperature and DO during some winters.

GOTM-FABM-PCLake is a one-dimensional model with focus on vertical ecological dynamics. The model suffices for most small lakes, but for lakes with large horizontal heterogeneity, a multi-dimensional model would be necessary. For example, the horizontal impact of wind is ignored in GOTM-FABM-PCLake, as the model assumes that there are no biogeochemical gradients in the

horizontal direction—although this can have important influence on the distribution of surface algal blooms. Most ecological systems reveal heterogeneity and patchiness on a broad range of scales [35,36], and this patchiness can be fundamental to population dynamics, community organization, and stability.

The focal point of our study was the direct consequences of heatwaves and their frequency on lake ecosystems. However, heatwaves may also affect processes within the watershed and consequently alter loads of water and nutrients into the lake. The heatwave of summer 2018 not only elevated temperature but also diminished rain (the driest year since 1996 [2]), leading to drought and consequently drops in productivity in the arable land. Such altered conditions within the watershed may carry memory between growing seasons potentially beyond what was found through our study design. Employing both watershed modeling and lake modeling could provide additional insight into the consequences of heatwaves on lake ecosystems.

## 4.4. Implications for Lake Management

Previous model studies on the impact of extreme climate events on aquatic ecosystems are sparse. Some of the studies that have been conducted to predict the effects on lake ecosystems [36,37] showed that an extreme climate leads to lake water quality deterioration. The methods applied in these former model studies have largely been simple or based on uncalibrated models, which only provide a potential qualitative tendency of the development of water quality in extreme climate scenarios rather than quantitative estimates. Here, we used a calibrated complex ecological model to predict the impacts of extreme climate events on a lake ecosystem. The model results indicate that the future extreme climate will lead to enhanced summer average Chl a concentrations and cyanobacteria biomass with a concurrent increase in summer algal blooms and their duration. The peak Chl a concentrations increased up to 46% compared with the baseline, and the timing of the bloom period occurred up to half a month earlier than normal and lasted longer. In addition, we found that the effects of extreme warming generally diminished within one or two months following the event, which was similar to the time of hydraulic retention. The variations of summer average TN concentrations and TP concentrations under the impacts of extreme climate were not pronounced, and the largest changes of TN concentrations and TP concentrations were 13% and 3.5%, respectively. Thus, lake managers will be challenged with additional problems in order to obtain good ecological status and water quality in a future with expectations of more frequent heat waves. On the positive side, our results indicate that the capacity of the lakes to remove nitrogen might increase, whereby the lakes become a better filter for nitrogen in transport from land to the sea.

**Author Contributions:** W.C., F.H., M.S. and E.J. wrote review and edited; W.C., Q.C., F.H., T.K.A., A.N. and D.T. helped to calibrate and validate; W.C., A.N., T.K.A. and D.T. collected data, visualized result and discussed. All authors have read and agreed to the published version of the manuscript.

**Funding:** This work was partly supported by a scholarship awarded to W.Y. Chen from the China Scholarship Council (Grant No. 201706710085) enabling a visit to Aarhus University. T.K. Andersen was supported by Ph.D. funding from the Sino-Danish Center for Education and Research. D. Trolle was supported through the CASHFISH project funded by the Danish Council for Independent Research. D. Trolle, A. Nielsen and E. Jeppesen was also supported by the WATExR project, funded through the EU JPI Climate initiative, the PROGNOS project, funded through the EU JPI Water initiative, and a project on Mechanistic Models for Water Action Planning funding by the Danish EPA.

**Acknowledgments:** We are grateful to A. M. Poulsen for manuscript editing.

**Conflicts of Interest:** The authors declare no conflict of interest.

## Appendix A

| Parameter | Description | Unit | Parameter Value | |
|---|---|---|---|---|
| | | | **Default** | **Adjusted** |
| **Abiotic_Water Module** | | | | |
| cThetaAer | temperature coefficient for reaeration | [−] | 1.024 | 1.005 |
| cThetaNitr | temperature coefficient for nitrification | [−] | 1.08 | 1.003 |
| cVSetPOM | maximum settling rate of POM | $m \cdot day^{-1}$ | −0.25 | −0.35 |
| cVSetIM | maximum settling rate of inorganic matter | $m \cdot day^{-1}$ | −1.0 | −0.95 |
| hNO3Denit | quadratic half saturation $NO_3$ conc. for denitrification | $mgN \cdot L^{-1}$ | 2.0 | 0.3 |
| hO2BOD | half saturation oxygen conc. for BOD | $mgO_2 \cdot L^{-1}$ | 1.0 | 4.79 |
| hO2Nitr | half saturation oxygen conc. for nitrification | $mgO_2 \cdot L^{-1}$ | 2.0 | 3.57 |
| kNitrW | nitrification rate constant in water | $day^{-1}$ | 0.1 | 0.39 |
| NO3PerC | denitrified $NO_3$ per mol C mineralised | mol $NO_3$ | 0.8 | 1.47 |
| O2PerNH4 | used $O_2$ per mol $NH_4+$ nitrified | mol $O_2$ | 2.0 | 3.30 |
| cThetaMinPOMW | temperature coefficient for mineralization from POM to DOM | [−] | 1.07 | 1.01 |
| kDMinPOMW | decomposition constant for POM-DW to DOM-DW | $day^{-1}$ | 0.01 | 0.0001 |
| kNMinPOMW | decomposition constant for POM-N to DOM-N | $day^{-1}$ | 0.01 | 0.014 |
| kPMinPOMW | decomposition constant for POM-P to DOM-P | $day^{-1}$ | 0.01 | 0.0003 |
| kSiMinPOMW | decomposition constant for POM-Si to DOM-Si | $day^{-1}$ | 0.01 | 0.0097 |
| cThetaMinDOMW | temperature coefficient for DOM mineralization | [−] | 1.07 | 1.05 |
| kDMinDOMW | mineralization constant of dissolved organic DW | $day^{-1}$ | 0.01 | 0.035 |
| kNMinDOMW | mineralization constant of dissolved organic N | $day^{-1}$ | 0.01 | 0.016 |
| kPMinDOMW | mineralization constant of dissolved organic P | $day^{-1}$ | 0.01 | 0.014 |
| kSiMinDOMW | mineralization constant of dissolved organic Si | $day^{-1}$ | 0.01 | 0.0083 |
| **Abiotic_Sediment Module** | | | | |
| fRefrPOMS | refractory fraction of sediment POM | [−] | 0.15 | 0.08 |
| O2PerNH4 | $O_2$ used per mol $NH_4+$ nitrified | mol | 2.0 | 1.71 |
| kNitrS | nitrification rate constant | $day^{-1}$ | 1.0 | 0.34 |
| cThetaNitr | temperature coefficient for nitrification | [−] | 1.08 | 1.01 |
| NO3PerC | $NO_3$ denitrified per mol C mineralised | [−] | 0.8 | 0.91 |
| hNO3Denit | quadratic half-sat. $NO_3$ conc. for denitrification | $mgN \cdot L^{-1}$ | 2.0 | 0.25 |
| kPSorp | P sorption rate constant not too high -> model speed | $day^{-1}$ | 0.05 | 0.089 |
| cRelPAdsD | max. P adsorption per g DW | $gP \cdot gD^{-1}$ | $3 \times 10^{-5}$ | $4.76 \times 10^{-5}$ |
| cRelPAdsFe | max. P adsorption per g Fe | $gP \cdot gFe^{-1}$ | 0.065 | 0.055 |
| fFeDIM | Fe content of inorganic. matter | $gFe \cdot Gd^{-1}$ | 0.01 | 0.026 |
| fRedMax | max. reduction factor of P adsorption affinity | [−] | 0.9 | 0.86 |
| cKPAdsOx | P adsorption affinity at oxidized conditions | $m^3 \cdot gP^{-1}$ | 0.6 | 1.6 |
| coPO4Max | max. SRP conc. in pore water | $mgP \cdot L^{-1}$ | 1.0 | 1.04 |
| cThetaDif | temperature coefficient for diffusion | [−] | 1.02 | 1.02 |
| kNDifNH4 | molecular $NH_4$ diffusion constant | $m^2 \cdot day^{-1}$ | 0.000112 | 0.000112 |
| cTurbDifNut | bioturbation factor for diffusion | [−] | 5.0 | 14.76 |
| kO2Dif | molecular $O_2$ diffusion constant | $m^2 \cdot day^{-1}$ | $2.6 \times 10^{-5}$ | 0.00018 |
| cTurbDifO2 | bioturbation factor for diffusion | [−] | 5.0 | 1.93 |
| kDMinHum | maximum decomposition constant of humic material ($1D^{-5}$) | $day^{-1}$ | $1 \times 10^{-5}$ | 0.00021 |
| cThetaMinPOMS | temperature coeff. for sediment mineralization of POM to DOM | [−] | 1.07 | 1.05 |
| kDMinPOMS | mineralization constant in sediment from POM-DW to DOM-DW | $day^{-1}$ | 0.002 | 0.0027 |
| kNMinPOMS | mineralization constant in sediment from POM-N to DOM-N | $day^{-1}$ | 0.002 | 0.0002 |
| kPMinPOMS | mineralization constant in sediment from POM-P to DOM-P | $day^{-1}$ | 0.002 | 0.0001 |
| cThetaMinDOMS | exp. temperature constant of sediment mineralization | [−] | 1.07 | 1.02 |
| kDMinDOMS | mineralization constant for sediment dissolved organic matter | $day^{-1}$ | 0.002 | 0.0027 |
| kNMinDOMS | mineralization constant for sediment dissolved organic N | $day^{-1}$ | 0.002 | 0.0017 |
| kPMinDOMS | mineralization constant for sediment dissolved organic P | $day^{-1}$ | 0.002 | 0.0022 |
| kSiMinDOMS | mineralization constant for sediment dissolved organic Si | $day^{-1}$ | 0.002 | 0.0019 |
| kDDifDOM | molecular diffusion constant for dissolved organic matter | $m^2 \cdot day^{-1}$ | 0.000112 | 0.0029 |
| kNDifDOM | molecular diffusion constant for dissolved organic N | $m^2 \cdot day^{-1}$ | 0.000112 | 0.000118 |
| kPDifDOM | molecular diffusion constant for dissolved organic P | $m^2 \cdot day^{-1}$ | 0.000112 | 0.000118 |
| sPAIMS | sediment absorbed phosphate | $g \cdot m^{-2}$ | 2.0 | 0.5 |
| sDPOMS | sediment particulate organic DW | $g \cdot m^{-2}$ | 474 | 1104 |
| sNPOMS | sediment particulate organic N | $g \cdot m^{-2}$ | 6.0 | 15.13 |
| sPPOMS | sediment particulate organic | $g \cdot m^{-2}$ | 1.0 | 10 |
| sDHumS | sediment humus DW | $g \cdot m^{-2}$ | 3719 | 4488 |

| Parameter | Description | Unit | Parameter Value | |
|---|---|---|---|---|
| | | | Default | Adjusted |
| **Phytoplankton_Water Module** | | | | |
| cSigTmDiat | temperature constant diatoms (sigma in Gaussian curve) | °C | 20.0 | 15.66 |
| cTmOptDiat | optimum temperature of diatoms | °C | 18.0 | 20.29 |
| cSigTmBlue | temperature constant blue-greens (sigma in Gaussian curve) | °C | 12.0 | 12.16 |
| cTmOptBlue | optimum temperature of blue-greens | °C | 25.0 | 28.11 |
| cSigTmGren | temperature constant greens (sigma in Gaussian curve) | °C | 15.0 | 12.12 |
| cTmOptGren | optimum temperature of greens | °C | 25.0 | 19.09 |
| cPDDiatMin | minimum P/DW ratio for diatoms | mg P·mg DW$^{-1}$ | 0.0005 | 0.0024 |
| cNDDiatMin | minimum N/DW ratio for diatoms | mg N·mg DW$^{-1}$ | 0.01 | 0.005 |
| cPDGrenMin | minimum P/DW ratio greens | mg P·mg DW$^{-1}$ | 0.0015 | 0.0018 |
| cNDGrenMin | minimum N/DW ratio greens | mg N·mg DW$^{-1}$ | 0.02 | 0.013 |
| cPDBlueMin | minimum P/DW ratio blue-greens | mg P·mg DW$^{-1}$ | 0.0025 | 0.0012 |
| cNDBlueMin | minimum N/DW ratio blue-greens | mg N·mg DW$^{-1}$ | 0.03 | 0.018 |
| cLOptRefDiat | optimum PAR for diatoms at 20 °C | W·m$^{-2}$ | 54.0 | 23.32 |
| cLOptRefGren | optimum PAR for greens at 20 °C | W·m$^{-2}$ | 30.0 | 35.82 |
| cLOptRefBlue | optimum PAR for blue-greens at 20 °C | W·m$^{-2}$ | 13.6 | 13 |
| cMuMaxBlue | maximum growth rate blue-greens | day$^{-1}$ | 0.6 | 1.38 |
| cMuMaxGren | maximum growth rate greens | day$^{-1}$ | 1.5 | 1.13 |
| cMuMaxDiat | maximum growth rate diatoms | day$^{-1}$ | 2.0 | 3 |
| kMortBlueW | mortality constant of blue-greens in water | day$^{-1}$ | 0.01 | 0.15 |
| cVNUptMaxDiat | maximum N uptake capacity of diatoms | mg N·mg DW$^{-1}$·day$^{-1}$ | 0.07 | 0.067 |
| cVNUptMaxBlue | maximum N uptake capacity of blue-greens | mg N·mg DW$^{-1}$·day$^{-1}$ | 0.07 | 0.079 |
| cAffNUptDiat | initial N uptake affinity diatoms | mg DW$^{-1}$·day$^{-1}$ | 0.2 | 0.19 |
| cAffNUptBlue | initial N uptake affinity bluegreens | mg DW$^{-1}$·day$^{-1}$ | 0.2 | 0.15 |
| fDissMortPhyt | soluble nutrient fraction of died algae | [−] | 0.2 | 0.41 |
| cVSetDiat | settling rate of diatoms | m day$^{-1}$ | −0.5 | −0.1 |
| cVSetGren | settling rate of greens | m day$^{-1}$ | −0.2 | −0.42 |
| cVSetBlue | settling rate of blue-greens | m·day$^{-1}$ | 0.06 | 0.03 |
| cChDBlueMax | maximum chlorophyll/C ratio for blue-greens | mg Chl a·mg DW$^{-1}$ | 0.015 | 0.0031 |
| cChDBlueMin | minimum chlorophyll/C ratio for blue-greens | mg Chl a·mg DW$^{-1}$ | 0.005 | 0.014 |
| cChDDiatMax | maximum chlorophyll/C ratio for diatoms | mg Chl a·mg DW$^{-1}$ | 0.012 | 0.021 |
| cChDDiatMin | minimum chlorophyll/C ratio for diatoms | mg Chl a·mg DW$^{-1}$ | 0.004 | 0.010 |
| cChDGrenMax | maximum chlorophyll/C ratio for greens | mg Chl a·mg DW$^{-1}$ | 0.02 | 0.020 |
| cChDGrenMin | minimum chlorophyll/C ratio for greens | mg Chl a·mg DW$^{-1}$ | 0.01 | 0.0075 |
| fPrimDOMW | fraction of dissolved organic matter from water column phytoplankton | [−] | 0.5 | 0.02 |
| kDRespBlue | maintenance respiration constant blue-greens | day$^{-1}$ | 0.03 | 0.047 |
| **Phytoplankton_Sediment Module** | | | | |
| fDissMortPhyt | soluble nutrient fraction of died algae | [−] | 0.2 | 0.01 |
| **Macrophytes Module** | | | | |
| cDVegIn | external macrophytes density | g D·m$^2$ | 1.0 | 0.46 |
| kMigrVeg | macrophyte migration rate | day$^{-1}$ | 0.001 | 0.0016 |
| cMuMaxVeg | maximum growth rate of macrophytes at 20 degrees | day$^{-1}$ | 0.2 | 0.031 |
| cDCarrVeg | maximum macrophytes standing crop | g DW·m$^{-2}$ | 400.0 | 207.78 |
| cDayWinVeg | day of the year for the end of growing season | day of the year | 259.0 | 218.90 |
| cTmInitVeg | temperature for onset of initial growth | °C | 9.0 | 10.08 |
| cCovSpVeg | specific cover | Gdw$^{-1}$·m$^{-2}$ | 0.5 | 0.27 |
| hLRefVeg | half-saturation for influence of light on macrophytes | W·m$^{-2}$ PAR | 17.0 | 18.05 |
| fWinVeg | fraction surviving in winter ([−], default = 0.3) | [−] | 0.3 | 0.31 |
| fSedUptVegMax | maximum sediment fraction of nutrient uptake | [−] | 0.998 | 0.64 |
| cHeightVeg | macrophytes height | m | 1.0 | 0.95 |
| cExtSpVeg | specific extinction of macrophytes | m$^2$·g DW | 0.01 | 0.0043 |
| cDVegMin | minimum dry weight of macrophytes in system | g DW·m$^{-2}$ | $1 \times 10^{-5}$ | $5.2 \times 10^{-5}$ |
| cQ10ProdVeg | temperature quotient of production | [−] | 1.2 | 1.18 |
| cQ10RespVeg | temperature quotient of respiration | [−] | 2.0 | 1.98 |
| **Zooplankton Module** | | | | |
| cTmOptZoo | optimum temperature for zooplankton | °C | 25.0 | 17.91 |
| kDRespZoo | maintenance respiration constant for zooplankton | day$^{-1}$ | 0.15 | 0.02 |
| cPrefDiat | selection factor for diatoms | [−] | 0.75 | 0.90 |
| cPrefBlue | selection factor for blue-greens | [−] | 0.125 | 0.25 |
| cPrefPOM | selection factor for particulate organic matter | [−] | 0.25 | 0.16 |
| hFilt | half-saturation constant for food conc. on zooplankton | g DW·m$^{-3}$ | 1.0 | 1.19 |
| fDAssZoo | dry weight assimilation efficiency of zooplankton | [−] | 0.35 | 0.33 |
| cFiltMax | maximum filtering rate | ltr·mg DW$^{-1}$·day$^{-1}$ | 4.5 | 1.11 |
| fZooDOMW | dissolved organic fraction from zooplankton | [−] | 0.5 | 0.36 |

| Parameter | Description | Unit | Parameter Value | |
|---|---|---|---|---|
| | | | **Default** | **Adjusted** |
| **Fish Module** | | | | |
| kDAssFiJv | maximum assimilation rate of zooplanktivorous fish | $day^{-1}$ | 0.12 | 0.121 |
| cDCarrPiscMax | maximum carrying capacity of piscivorous fish | $g\,DW{\cdot}m^{-2}$ | 1.2 | 2.74 |
| cCovVegMin | minimum submerged macrophytes coverage for piscivorous fish | % | 40.0 | 25.67 |
| hDVegPisc | half-saturation constant for macrophytes on piscivorous fish | $g{\cdot}m^{-2}$ | 5.0 | 2.08 |
| **Zoobenthos Module** | | | | |
| fBenDOMS | dissolved organic fraction from zoobenthos | [−] | 0.5 | 0.52 |
| **Auxiliary Module** | | | | |
| kVegResus | relative resuspension reduction per gram macrophytes | $m^2{\cdot}g\,DW^{-1}$ | 0.01 | 0.05 |
| kTurbFish | relative resuspension by adult fish browsing | $g{\cdot}gfish^{-1}{\cdot}day^{-1}$ | 1.0 | 2.15 |
| cVSedPOM | maximum sedimentation velocity of POM | $m{\cdot}day^{-1}$ | 0.25 | 0.5 |
| cVSedDiat | sedimentation velocity of diatoms | $m{\cdot}day^{-1}$ | 0.5 | 0.68 |
| cVSedGren | sedimentation velocity of greens | $m{\cdot}day^{-1}$ | 0.2 | 0.54 |
| cVSedBlue | sedimentation velocity of blue-greens | $m{\cdot}day^{-1}$ | 0.06 | 0.01 |
| crt_shear | critical shear stress | $N{\cdot}m^{-2}$ | 0.005 | 0.014 |

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
