# Peer review of "Modeling the Ecological Response of a Temporarily Summer-Stratified Lake to Extreme Heatwaves"

_water, doi:10.3390/w12010094_

Round 1

Reviewer 1 Report

Manuscript number: water-619525

Manuscript title:  Modelling the ecological response of a temporarily summer-stratified lake to extreme heatwaves

The paper presents very interesting results of  using ecosystem model GOTM-FABM-PCLake to determine the response of a shallow eutrophic lake to extreme climate events, such as heat wave year.

The manuscript fits with the journal aims. However, some information and points should be added before publication of the manuscript.

Title

If the term “ecological response” is adequate when You analyse only cyanobacteria biomass?

Methods

Study site

How the lake for this model was chosen, please specify

Please give some more data (mean values or ranges) of chl-a, cyanobacteria, TP and TN typical for the studied lake. It would be useful for potential reader to compare with other temperate lakes.

Results

Figure 2 is illegible

Caption of Figure 3 does not fully explain the results presented

Reviewer 2 Report

Review

Paper ID: water-619525

Modelling the ecological response of a temporarily summer-stratified lake to extreme heatwaves

Authors tried to determine the ecosystem response to extreme climate events for a temporarily stratified, eutrophic lake Lake Bryru in Denmark by using an ecosystem model (GOTM-FABM-PCLake) and air temperature data from the remarkably warm summer of 2018 as a basis for the extreme scenario analysis. They found modest impacts of the extreme climate events on nutrient levels, which in some scenarios decreased. The most significant impacts were found for phytoplankton, where summer average chlorophyll a concentrations and cyanobacteria biomass peaks were up to 39% and 58% higher than during baseline, respectively.

The topic is interesting and relevant to the water journal but not in this format. The quality of figure is low, the methodology well not connected with results and discussion. There is no conclusion for the paper.

Some other comments:

Figure 1. Need to improve the colors in the legend in not much with Map. For example I could not see any Yellow color in the map as agriculture while in text was mentioned “large amount of nutrients via inlets that drains agricultural areas”

I recommend the basic component of lake water balance will be added in the study area description.

Line 87 What did you mean Measures, the sentence is vague please rephrase it.

Please briefly quantify the contribution of different type land uses.

Please add a flowchart for Model description how GOTM , PCLake and FABM connected, clearly show input, output of each.

Please present input data graphically and divide to calibration and validation period,

Line 132 What is the warm-up period, please explain it in Model description

Please clearly addressed or reference all code and software that you used.

The Model calibration and validation (Part 2-4) is not clear please explain more detail. I think the authors could not explain clearly, what they did in this part.

The Scenarios need to be justified.

The Quality and layout of Figure 2 is low need to be modified.

Line 213 Why 6-17% please justify it.

Figure 6. Different panels need to be adjusted

Overally results were not well supported by methodology.

In discussion mentioned the focal point of study was the direct consequences of heatwave and their frequency on lake ecosystem while I did not see any supportive information in this regards in results or methodology.

Introduction need to be improved with wider point of view in lake literature, I recommend look at following literature:

Mechanisms and assessment of water eutrophication Analysis of Effective Environmental Flow Release Strategies for Lake Urmia Restoration Design of environmental flow regimes to maintain lakes and wetlands in regions with high seasonal irrigation demand. Determination of the distribution of phytoplankton related to water quality in sea bass (Dicentrarchus labrax) FARMING LAND PONDS (MuÄŸla, Turkey) Can lake sensitivity to desiccation be predicted from lake geometry? Effects of aquatic macrophytes on water quality and phytoplankton communities in shallow lakes A sensitivity analysis of lake water level response to changes in climate and river regimes
